# Charge-Transfer Induced by the Oxygen Vacancy Defects in the Ag/MoO_3_ Composite System

**DOI:** 10.3390/nano11051292

**Published:** 2021-05-14

**Authors:** Qi Chu, Jingmeng Li, Sila Jin, Shuang Guo, Eungyeong Park, Jiku Wang, Lei Chen, Young Mee Jung

**Affiliations:** 1College of Chemistry, Jilin Normal University, Siping 136000, China; cq325626@163.com; 2School of Public Health and Basic Medicine, The Chinese University of Hong Kong, Hong Kong 999077, China; lijingmengapp@163.com; 3Department of Chemistry, Institute for Molecular Science and Fusion Technology, Kangwon National University, Chunchon 24341, Korea; jsira@kangwon.ac.kr (S.J.); guoshuang@kangwon.ac.kr (S.G.); egpark@kangwon.ac.kr (E.P.)

**Keywords:** SERS, oxygen vacancy defects, Ag/MoO_3_ composite, charge-transfer

## Abstract

In this paper, an Ag/MoO_3_ composite system was cosputtered by Ar plasma bombardment on a polystyrene (PS) colloidal microsphere array. The MoO_3_ formed by this method contained abundant oxygen vacancy defects, which provided a channel for charge transfer in the system and compensated for the wide band gap of MoO_3_. Various characterization methods strongly demonstrated the existence of oxygen vacancy defects and detected the properties of oxygen vacancies. 4-Aminothiophenol (p-aminothiophenol, PATP) was used as a candidate surface-enhanced Raman scattering (SERS) probe molecule to evaluate the contribution of the oxygen vacancy defects in the Ag/MoO_3_ composite system. Interestingly, oxygen vacancy defects are a kind of charge channel, and their powerful effect is fully reflected in their SERS spectra. Increasing the number of charge channels and increasing the utilization rate of the channels caused the frequency of SERS characteristic peaks to shift. This interesting phenomenon opens up a new horizon for the study of SERS in oxygen-containing semiconductors and provides a powerful reference for the study of PATP.

## 1. Introduction

Surface-enhanced Raman scattering (SERS) has attracted much attention in the detection of biomolecules, chemicals, composite materials, environmental pollutants, etc. [1,2,3,4,5] and is a simple, fast, and sensitive technology. SERS phenomena were first observed on the surface of noble metals such as Ag, Au, and Cu, and noble metals are the first-choice materials for the early preparation of SERS substrates [6,7]. In 1974, Fleischman et al. was initially discovered SERS phenomenon on the rough silver electrode. Then, Van Duyne, Albrecht, and Creighton et al. proposed that some of the enhancement in the Raman scattering signal of the probe molecule comes from the surface enhancement of the rough silver electrode, and named this effect as surface-enhanced Raman scattering [8,9,10]. Due to the poor compatibility of noble metal materials, the adsorption sites of molecules on the surface of traditional noble metal SERS active materials are not uniform, and adsorption denaturation and other problems occur [11]. When a laser irradiates a noble metal, strong plasmon resonance and plasmon-driven catalysis reactions are induced, resulting in the poor reproducibility of SERS spectra [12]. Recent studies have shown that laser-induced inelastic scattering occurs on the surface of semiconductor thin films, leading to changes in the magnetic anisotropy and other properties of the semiconductor surface. These studies have proposed a method to improve the performance of semiconductors using magnons [13,14]. Raman scattering is also a kind of inelastic scattering. When this effect occurs on the surface of a semiconductor film, it will also cause changes in the distribution of electrons on the surface of the semiconductor. In recent years, there have been many studies on semiconductors exhibiting SERS activity [15,16,17]. By controlling the geometric structure, the electron density, the crystal morphology, and other semiconductor parameters, the ability for semiconductors to recognize target molecules can be enhanced, leading to better SERS performance. However, the low carrier concentrations on semiconductor surfaces result in weak surface plasmon resonance (SPR) in the visible area. Therefore, the optimal SERS effect can be achieved by utilizing the noble metal-semiconductor system. By combining noble metals with semiconductors, the SERS effect can be optimized, as the noble metals contribute electromagnetic (EM) enhancement, and the chemical enhancement (CM) of the system is stimulated. This approach has developed into a mature SERS research method [18,19,20]. 

In recent years, there has been some discussion on composite substrates composed of, for example, noble/noble metals (Au/Ag) and semiconductors (TiO_2_, ZnO, Cu_2_O, etc.). Based on previous reports, possible charge transfer (CT) paths in various structures were obtained. In substrates composed of noble metals and semiconductors, there are several assemblies that can form in the presence of probe molecules: metal-semiconductor-probe molecule, metal-probe molecule-semiconductor, and semiconductor-metal-probe molecule [21,22,23,24]. Yang et al. employed the self-assembly method using Au and ZnO as a composite substrate, and 4-aminothiophenol (p-aminothiophenol, PATP) was selected as the probe molecule to obtain the Au-ZnO-PATP system. In this system, the addition of ZnO significantly improved the enhancement effect. It was also confirmed that CT in the system, that is, through ZnO, involved charge carriers moving from Au to PATP [23]. Ji et al., using p-mercaptothiophenol (MPH) as a probe molecule and the noble metal Ag and semiconductor TiO_2_ as a composite substrate, assembled a Ag-MPH-TiO_2_ system [25]. They found that Ag provided a strong EM contribution in the system and that the introduction of the semiconductor TiO_2_ changed the CM proportion. The results showed that in the Ag-MPH system, charge was transferred from MPH to Ag, and electron transfer from Ag to MPH and from MPH to TiO_2_ occurred after TiO_2_ was introduced, resulting in the overall transition of electrons from Ag to TiO_2_. Jiang et al. synthesized a TiO_2_-p-mercaptobenzoic acid (MBA)-Ag structure. The results confirmed that in such a structure, TiO_2_ generates a surface state energy level after being excited by a laser. This energy level caused CT between TiO_2_ and MBA, and the introduction of Ag produced a relatively obvious EM contribution; as such, the SERS effect of the MBA molecule was significantly enhanced [26]. Therefore, semiconductors play an important role in the SERS of composite systems.

Molybdenum trioxide (MoO_3_) has many unique properties and exhibits outstanding performance in electrochromic materials, photochromic materials, photocatalysis, and biosensing [26,27,28,29]. MoO_3_ has also received much attention in the field of SERS [30,31,32]. Prabhu et al. synthesized MoO_3_ with a sea urchin morphology and obtained improved SERS results. [31] Zhu et al. doped MoO_3_ with hydrogen under mild conditions, and the carrier concentration on the surface of MoO_3_ was able to reach the concentration level of noble metals. [30] In addition, MoO_3_ with abundant oxygen vacancies and a specific structure has facilitated the adsorption of SERS probe molecules. Therefore, MoO_3_ has very broad prospects in the SERS field.

Previously, we reported the Ag-ZnSe-PATP system, which is the combination of noble metals and semiconductors. In Ag-ZnSe-PATP system, Ag and ZnSe were layer-by-layer sputtered on the polystyrene (PS) template. The proposed Ag-ZnSe composites compensate the CT difficulty in wide band gap semiconductors, which was initiated by the SPR of Ag [33]. In this study, we designed a Ag-MoO_3_-PATP system and performed various characterizations and SERS analyses. Ag and MoO_3_ thin films were deposited on PS colloidal microspheres by magnetron sputtering under the bombardment of an Ar plasma. The structure and characteristics of MoO_3_ formed by this method do not change significantly; however, it does contain abundant oxygen vacancies. This is because the high-energy Ar plasma selectively activates oxygen atoms, causing the oxygen atoms to diffuse from the bulk of the oxygen-containing semiconductor to its surface. Previous studies have confirmed that oxygen vacancies form when an Ar plasma bombards oxygen-containing semiconductor films. The effect of the obtained defect energy level on the generation of electrons and holes has been confirmed by theoretical calculations [34]. In our research, bombarding the Ag-MoO_3_ system with an Ar plasma also produced a substrate rich in oxygen vacancies, which was confirmed by transmission electron microscopy (TEM) and X-ray photoelectron spectroscopy (XPS) results. The presence of oxygen vacancies in the Ag-MoO_3_ system strengthens CT [16] between the two materials and is reflected in the SERS spectrum of PATP, resulting in band positions that have not been assigned in previous studies.

## 2. Materials and Methods

### 2.1. Materials

Ag targets (99.99%) were purchased from Beijing Hezong Tianrui Technology Development Center. MoO_3_ (99.99%) target materials were purchased from Beijing Jingmaiyan Material Technology Co., Ltd. (Beijing, China). PS colloidal microspheres with 200 nm diameters were purchased from Sinopharm Chemical Reagent Co., Ltd (Shanghai, China). PATP was purchased from Aladdin Reagent Co., Ltd. (Shanghai, China). Anhydrous ethanol was purchased from Sinopharm Chemical Reagent Co., Ltd. (Shanghai, China). All chemical reagents were used without further purification.

### 2.2. Preparation of the Ag/MoO_3_ Two-Dimensional Arrays

The preparation process for the Ag/MoO_3_ two-dimensional arrays is shown in Scheme 1. The PS microspheres were diluted with an equal volume of absolute ethanol, and the diluted microspheres were dispersed on the surface of a washed silicon wafer (10 cm × 5 cm). The silicon wafer was then placed in water so that the surface of PS microspheres remained on the water surface. A second washed silicon wafer (2 cm × 2 cm) were placed in the water and were gently picked up to obtain the monolayer of PS microspheres. The PS microspheres on the water surface were removed to obtain a uniform microsphere film, and the film was dried in air for later use. In the magnetron sputtering system (ATC 1800-F, AJA, Scituate, MA, USA), the target materials were bombarded with Ar gas, and Ag and MoO_3_ were simultaneously sputtered on the prepared PS array. The Ag sputtering power was 10 W, the MoO_3_ sputtering power was 50, 70, or 90 W, and the sputtering time was 15 min. In addition, a sample was prepared by only sputtering Ag, and the sputtering power and sputtering time were 10 W and 15 min, respectively. A sample that was only sputtered with MoO_3_ was also prepared, and the sputtering power and sputtering time were 90 W and 15 min, respectively. The working pressure of the system was 6 × 10^−1^ Pa, and the Ar flow rate was 9 sccm (standard cubic centimeters per minute). For SERS measurements, PATP was dissolved in absolute ethanol to prepare a solution with a concentration of 10^−3^ M. All samples were soaked with the PATP solution for 5 h and then washed with absolute ethanol three times.

### 2.3. Characterization of the Ag/MoO_3_ Two-Dimensional Arrays

The morphologies of the different Ag/MoO_3_ samples were observed with scanning electron microscopy (SEM, JEOL 6500F, JEOL, Tokyo, Japan) using a microscope operated at a voltage of 200 kV. Transmission electron microscopy (TEM, JEM-2100F, JEOL, Tokyo, Japan) was used to obtain high-resolution images of the samples under a 200 kV voltage. Ultraviolet-visible-near infrared (UV-Vis-NIR) absorption spectra were obtained using a Shimadzu UV-3600 spectrophotometer (Shimadzu, Kyoto, Japan). A Raman spectrometer (Renishaw Raman System Confocal 2000 Microphotometer, Renishaw, London, UK) with a 514 nm excitation light source equipped with a CCD detector and a holographic notch filter was used to obtain SERS spectra. The laser was focused on the surface of the sample through a 50× long-distance objective lens with a 1 µm spot size. When the SERS experiment was performed, the laser power employed was controlled from 100 to 20 mW. PATP is used as a probe molecule and is formulated into an absolute ethanol solution with a concentration of 10^−3^ M. The collection time was 20 s, and the number of acquisitions was 1. To determine the crystal structures of the samples, each sample was characterized by X-ray diffraction (XRD) using a Rigaku-MiniFlex600 system (Rigaku, Tokyo, Japan). To determine the element and its state, X-ray photoelectron spectroscopy (XPS) spectra of the sample were obtained with a Thermo-Scientific-Escalab 250 XI Al440 system (Thermo Fisher Scientific, Waltham, MA, USA), and all results were corrected with carbon peaks (C 1s = 284.6 eV).

## 3. Results

### 3.1. Characterization of the Ag/MoO_3_-Coated PS Templates

As shown in Scheme 1, the self-assembly method was employed to prepare the PS microsphere arrays. The substrate was prepared by cosputtering Ag and MoO_3_. The topography of the prepared substrates with different doping ratios can be visually observed. The morphology of Ag and MoO_3_ on the cosputtered substrates was characterized by using SEM. Figure 1 shows the morphology of the Ag/MoO_3_ composites with different doping ratios. Figure 1a shows the sample sputtered with Ag, and the sputtering power and time were 10 W and 15 min, respectively. In the samples shown in Figure 1b–d, the Ag sputtering power was 10 W, the MoO_3_ sputtering power was 50, 70, or 90 W, and the sputtering time was 15 min. Figure 1e shows the PS array sputtered with MoO_3_, and the sputtering power and time were 90 W and 15 min, respectively. A PS array without sputtered Ag or MoO_3_ is shown in Figure 1f for comparison. The elemental distribution of the samples is shown on the right side of the SEM images. With increasing MoO_3_ sputtering power from 50 to 90 W, the gaps between the PS microspheres are gradually filled, and the surface of the sample tends to be smooth, which was confirmed by the TEM images in the upper right corner of each image. The changes in morphology observed in Figure 1 will affect the hot spot distribution of the substrate, thus affecting SERS characteristics [35].

Figure 2 shows high-resolution TEM images of the Ag/MoO_3_ composites. Figure 2a,b show that Ag and MoO_3_ form a hemispherical core-shell on the PS microspheres. Figure 2c shows the crystallization of Ag on the substrate, and the lattice spacing is 0.233 nm, which is assigned to the (220) plane of Ag. Figure 2d shows the crystallization of MoO_3_, and the lattice spacing is 0.231 nm, which is assigned to the (202) crystal plane of MoO_3_. Figure 2 shows that in the Ag/MoO_3_ system, both Ag and MoO_3_ crystal lattices exist. The distribution of Ag and MoO_3_ proves the uniformity of the system obtained by cosputtering. This method allows Ag and MoO_3_ particles to be embedded alternately without covering each other and has little interference with SERS performance. In addition, oxygen vacancy defects also exist on the surface of the system, which is very beneficial to the charge transition. Interestingly, in Figure 2d, we also see an amorphous region (the region between the two red lines) between the two lattice planes. Previously, researchers identified and proved that this corresponds to oxygen vacancy defects [36]. We speculate that the amorphous regions appearing in Figure 2d also correspond to oxygen vacancy defects, and XPS is used to prove this speculation by analyzing the elemental states.

The crystallization of the samples was determined from XRD spectra (Figure 3). Spectra a, b–d, and e in Figure 3 show the XRD spectra of the PS arrays sputtered with pure Ag, Ag-MoO_3_ using different MoO_3_ sputtering powers, and pure MoO_3_, respectively. With increasing MoO_3_ sputtering power, the intensity of the Ag diffraction peak decreases gradually. The diffraction peaks at 13.73°, 16.74°, 25.53°, 38.20°, 44.47°, 64.78°, and 77.82° correspond to the (110), (040), (111), (200), (220), (440), and (222) lattice planes of Ag, respectively. The diffraction peak at 69.39° corresponds to the (202) lattice plane of MoO_3_. The diffraction peak for MoO_3_ at 69.39° is sharp and strong, which indicates that the crystal is dominated by the (202) lattice plane. There is an impurity peak at 33.04° in spectrum e (Figure 3), which may be attributed to the amorphous region caused by oxygen vacancy defects. The XRD spectra show that with increasing MoO_3_ content, the intensity and sharpness of the Ag diffraction peaks on the sample surface become weaker. This is because Ag becomes increasingly covered by MoO_3_. The increase in MoO_3_ content also leads to the SPR of Ag weakening, which is consistent with the SERS results.

Figure 4 shows the XPS spectra of the Ag/MoO_3_ system. The spectrum shows the chemical composition and electronic structure of each sample. From the XPS spectra of Ag and Mo elements shown in Figure 4A,B, respectively, we found that the characteristic peaks of the elements of the composite material shifted compared with those of the individual elements. This is due to the polarization of the electron cloud in the composite system, and the binding energy of the elements has changed. Figure 4C shows two states of oxygen through peak fitting. The O_a_ peak marked by the red dashed line comes from the oxygen vacancies in the matrix [37]. We found that with the increase in MoO_3_ content, the peak attributed to oxygen vacancies gradually became obvious, and its intensity increased. The O_b_ peak comes from the lattice oxygen in MoO_3_, and its intensity gradually weakens. From the XPS spectrum of oxygen, we confirmed the existence of oxygen vacancies and found changes in the content of oxygen vacancies. The results obtained by XPS analysis clearly confirmed that the amorphous region in the TEM image was indeed caused by oxygen vacancies. In addition, we calculated the percentage of the elemental content in each sample based on the XPS test results (as shown in Figure 5). In the calculation, we employed the carbon peak for correction to deduce the interference of carbon element. With the increase in MoO_3_ content, we observed that the Ag content gradually decreases. Since the stoichiometric ratio of O to Mo in MoO_3_ is 3:1, the O content increases significantly, while the Mo content increases slightly.

Figure 6 shows the UV-Vis-NIR absorption spectra of the PS arrays sputtered with Ag, Ag/MoO_3_ and MoO_3_. The peak attributed to PS microspheres appears at approximately 340 nm. With the increase in MoO_3_ sputtering power, the content of Ag and MoO_3_ on the surface of the hemispherical shell is constantly changing, and the distribution of each component is irregular. Therefore, the peak attributed to PS microspheres may be coupled with the Ag peak, thus changing the intensity and position of this peak. For the spectrum a in Figure 6, the spectrum of the PS array sputtered with Ag shows a broad band at 448 nm, which is attributed to the Ag shells on the PS microsphere array. A broad absorption band corresponding to MoO_3_ appears at about 400–700 nm (spectrum e in Figure 6) [38]. With increasing MoO_3_ sputtering power, the absorption bands for MoO_3_ and Ag become coupled, leading to an increase in the band intensity. Additionally, a redshift is observed for the band assigned to the Ag shells (spectra b–d in Figure 6). Using PS microspheres with a diameter of 200 nm as a template, a hemisphere covered by Ag and MoO_3_ was formed alternately. The intensity of the dipole at the edge of this hemispherical structure is very large, even greater than that of the top dipole [39]. Such a hemisphere with an increasingly stronger electric field from the top to the edge is formed. The electron group near the spherical shell enhances not only the SERS signal but also the localized surface plasmon resonance (LSPR) of the composite substrate (see the red asterisk). By increasing the MoO_3_ content, a redshift is observed for the LSPR bands of the Ag/MoO_3_ composites, as shown by the red asterisk in spectra b–d (Figure 6). Since the increase in the number of oxygen vacancy defects is a microscopic and gradual change, the red shift in this band is not obvious. Based on previous reports on MoO_3_ [37,40,41], this is due to the existence of oxygen vacancy defects and their interaction with Ag. With an increase in the MoO_3_ content, the oxygen vacancy defects reduce the actual band gap of MoO_3_, CT between Ag and MoO_3_ occurs easily, and the photon energy required by the system decreases. The redshift of the absorption band at approximately 500 nm is dependent on the MoO_3_ sputtering power. The UV-Vis-NIR absorption spectra show changes in the optical properties, especially the LSPR. In addition, we accurately determined the excitation wavelength used in the SERS study, which was based on the position of the main absorption band in the UV-Vis-NIR absorption spectra.

### 3.2. SERS Characteristics of the Ag/MoO_3_-Coated PS Templates

PATP was employed as a probe molecule to evaluate the SERS properties of Ag/MoO_3_. The samples were soaked in a PATP solution with a concentration of 10^−3^ M to complete binding to PATP. SERS spectra of PATP adsorbed on the Ag, Ag/MoO_3_, and MoO_3_-coated PS arrays are shown in Figure 7A. The laser power used at this time was 20 mW. The SERS spectra of PATP adsorbed on the Ag/MoO_3_ composites exhibited unique changes when compared with the SERS spectrum of PATP adsorbed on the Ag-coated PS array. Two new obvious bands emerged at 1556 and 1168 cm^−1^ when the MoO_3_ sputtering power was increased. In addition, a blueshift in the band at 1305 cm^−1^ was detected. This occurred due to the CT contribution from the Ag/MoO_3_ composites. Therefore, the contribution of CT to SERS intensity was studied according to the equation used to describe the degree of CT (ρ_CT_), which was proposed by Lombardi and Birke [42]:ρ_CT_(k) = [I_k_(CT) − I_k_(SPR)]/[I_k_(CT) + I_0_(SPR)]
where k corresponds to a single molecule in the SERS spectrum, I_k_ (CT) corresponds to the Raman band enhanced by CT, and I_0_ (SPR) corresponds to the Raman band enhanced by EM. If ρ_CT_(k) is greater than 0.5, CT is mainly responsible for the SERS enhancement [41]. 

As shown in Figure 7A, the strongest SERS signal is observed when the MoO_3_ sputtering power is 50 W, which is the most conducive to SERS enhancement. Therefore, the ρ_CT_ equation was employed to evaluate the CT process. The ρ_CT_ of each sample was calculated with respect to the bands at 1434 and 1079 cm^−1^. The ρ_CT_ values for the Ag/MoO_3_ composites prepared with different MoO_3_ sputtering powers are shown in Figure 7A. 

As shown in the Figure 8A of the relationship between ρ_CT_ and the MoO_3_ sputtering power, the ρ_CT_ values for the Ag/MoO_3_ composites prepared with MoO_3_ sputtering powers of 0, 50, 70, and 90 W are 0.48, 0.70, 0.64, and 0.53, respectively. The observation of two new bands and a blueshifted band is due to CT between the Ag/MoO_3_ composites and PATP. The high ρ_CT_ is due to the oxygen vacancy defects in MoO_3_, which provide an electron channel, and the electrons are excited by the SPR of Ag. Through this new channel, the electrons jump to the energy level of PATP, leading to an electron cloud rearrangement and the formation of a new b_2_ mode. Although the structure of PATP is relatively simple, the physical and chemical activities of PATP are complex [43,44]. 

With increasing MoO_3_ sputtering power, the bands at 1168, 1331, and 1556 cm^−1^ become increasingly obvious. Combined with previously reported theoretical calculations on PATP [45], the band at 1168 cm^−1^ is attributed to the contribution of β(CH). The bands at 1331 and 1556 cm^−1^ are attributed to the contributions of β(CH) and υ(CC), respectively. The detailed band assignments are listed in Table 1. These new vibration modes are all related to the plane of the benzene ring. We believe that the abundant oxygen vacancy defects provide channels for electron movement. The large-scale “Ag/MoO_3_-PATP” electron migration not only results in the vibration of chemical bonds outside the benzene ring but also causes the chemical bonds in the plane of the benzene ring to vibrate. Due to the relatively stable structure of the benzene ring, this additional vibration can only occur when CT is very active in the system. With increasing MoO_3_ sputtering power, although the overall intensity of the SERS bands decreases, the positions of the bands at 1168, 1331, and 1556 cm^−1^ remain obvious, and their relative intensities increase, which confirms the effect of oxygen vacancy defects on the CT system. The CT mechanism is depicted in Figure 8B. 

To verify the importance of oxygen vacancy defects, the laser power-dependent experiment was performed. By increasing of the laser power from 20 to 100 mW (spectra a to i in Figure 7B), the intensities of bands at 1168 and 1556 cm^−1^ were significantly increased which indicated that the number of excited electrons was increased. Herein, the oxygen vacancy defect acts as an intermediate energy level (electron sink and the shallow donor level) and promotes the CT to the molecule (as shown in Figure 8B). Therefore, oxygen vacancy defect results in a broader donor energy level distribution, which provides the more effective passageways for electron transitions and compensates for the wide band gap of the MoO_3_ semiconductor [33]. Thus, the higher laser power, the more excited the electrons; the higher utilization rate of oxygen vacancy defects, the larger the scale of electronic transitions.

## 4. Conclusions

We successfully prepared a Ag/MoO_3_ composite system containing oxygen vacancy defects and performed a SERS study with PATP molecules. High-resolution TEM image analysis proved the existence of oxygen vacancy defects. In this system, the oxygen vacancy defects act as a charge channel to assist CT in wide band gap semiconductors. The SERS results show that the CT induced by oxygen vacancies results in the formation of a new b_2_ mode band, which confirms our proposed mechanism. This study provides a reference for future studies of SERS in oxygen-containing semiconductors and new band assignments for PATP. Therefore, it opens a new field for the SERS-based study of oxygen vacancy defect-containing semiconductors.

## Data Availability

Do not have the supporting reported results.

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
