# Peer review of "Charge-Transfer Induced by the Oxygen Vacancy Defects in the Ag/MoO3 Composite System"

_nanomaterials, 2021, doi:10.3390/nano11051292_

Round 1

Reviewer 1 Report

The resubmitted manuscript by Qi Chu and coworkers is an interesting paper describing technology of SERS substrate fabrication. The authors introduce method of SERS structure fabrication that combine semiconductors and noble metal. The structure exhibits very good SERS activity. The manuscript is very well written and according my opinion shows very promising results.

-In the Introduction (line 29), the authors should mention more general reference of SERS examples of biological and chemical targets resp. enhancement of Raman signal due to nanoparticles. And it has to be mentioned the first experiment on the nobel metal electrod (line 32). There are some useful references such as:

  • Bernatová SDonato M. GraziaJeĹľek JPilát ZSamek OMagazzu AMaragò OMZemánek PWavelength-Dependent Optical Force Aggregation of Gold Nanorods for SERS in a Microfluidic ChipJ. Phys. Chem. C123, 5608-5615 

  • Judith Langer et al. Present and Future of Surface-Enhanced Raman, ACS Nano 2020, 14, 1, 28–117 https://doi.org/10.1021/acsnano.9b04224

  • M. Fleischmann, P.J. Hendra, A.J. McQuillan, Raman spectra of pyridine adsorbed at a silver electrode, Chemical Physics Letters, Volume 26, Issue 2,1974, https://doi.org/10.1016/0009-2614(74)85388-1.

but feel free to select different citations .

-In the section Materials and Methods (line 138-139), the authors say:

..”When the SERS experiment was performed, the laser power employed was controlled from 100 to 20 mW. The collection time was 20 s, and the number of acquisitions was 1.” …

As a tested substance for SERS structure p-aminothiophenol (concentration probably 10-3 M) was used.

1)Please, comment limits of detection (LOD) of p-aminothiophenol for standard Raman analysis.

2)Line 275, missing unit of concentration (M, mg/L?).

In the section Results (278-280), the authors say:

...”The SERS spectra of PATP adsorbed on the Ag/MoO3 composites exhibited unique changes when compared with the SERS spectrum of PATP adsorbed on the Ag-coated PS array. Two new obvious bands emerged at 1556 and 1168 cm-1 when the MoO3 sputtering power was increased.”...

How did you eliminate the influence of Raman signal of the substrate? Please comment (or add) SERS spectrum of the substrate without the tested material (PATP).

It is clearly visible that authors added missing description and comments. I don't see any obstacle (after some minor comments) in accepting this manuscript as a paper to Nanomaterials.

Author Response

We appreciate the Reviewer’s comments on our manuscript, which have helped us significantly revise the manuscript.

Our responses to the comments of Reiewer is attahed.  

Reviewer 2 Report

Thank you for taking into account all my critical remarks and suitable changes of the manuscript. In my opinion all the changes have improved the text.  It was a good idea to add two figures, namely Fig. 4 and Fig. 5. I think that the revised manuscript can be accepted for publication in Nanomaterials. However, I have still some remarks:

  • Page 3, line 134. It should be UV-Vis-NiR spectra.
  • I am surprised that Raman measurements were performed using so high laser powers (100 – 20 mW) and that the number of acquisitions was only 1. I am wondering about stability of the samples, especially stability of PATP? It seems to me that it would be better to reduce the laser power and increase the number of acquisitions.
  • Page 11, figure caption, lines 332 and 333. The laser power units are “mW”.

Author Response

(The authors gave the same response as above.)

Author Response

(The authors gave the same response as above.)

Round 2

Reviewer 3 Report

The authors improved their manuscript following the received suggestions and the paper can now be accepted for publication.

This manuscript is a resubmission of an earlier submission. The following is a list of the peer review reports and author responses from that submission.

Round 1

Reviewer 1 Report

Surface-enhanced Raman scattering (SERS) is an important and very sensitive technique, used successfully in many practical applications. Originally, SERS was observed on surface of metals (usually Au or Ag) and their nanostructures. Recently, it was demonstrated that similar effects can be observed for substrates prepared of semiconductors, especially structures composed of semiconductors and noble metals. The reviewed paper belongs to this interesting direction of research. The authors have shown that oxygen vacancy defects exist and they play are an important role in the electron transfer. My remarks are as follows:

- Page 2, lines 48-52. We read that "Yang at al. employed the self-assembly method using Au and ZnO [...] to obtain Au-ZnO-PATP system. [...] charge carriers moving from au to PATP [16]." However, the Ref. [16] is about doped Ag–TiO2 hybrid. 

-  Page 2, lines 69-73. We read that “Zhu et al. doped MoO3 with hydrogen under mild conditions, and the carrier concentration on the surface of MoO3 was able to reach […] broad prospects in the SERS field. [25]” However, none of the earlier cited three references begins with “Zhu at al.” and the Ref. [25] which – as can be supposed – is a citation related to the above sentence is not about “doped MoO3 with hydrogen”.

-  Page 2, lines 79-80. We read that “…high-energy Ar plasma selectively activates oxygen atoms, causing the oxygen atoms to diffuse from the bulk of the oxygen-containing semiconductor to its surface. [26]”. However, the Ref. [26] is about “Argon (Ar) plasma treatment […] of indium tin oxide (ITO) thin films”. Are the authors sure that the effect observed for ITO thin films is analogous to that one in Ag-MO3-PAT? In my opinion this problem should be discussed in more detail.

-  The first sentences of the paragraph 3.1 are actually a repetition of information already given in the paragraph 2.2.

-  Page 4, lines 149-150. I wonder why “…the surface tends to be smooth because the particle diameter of MoO3 is smaller than that of Ag.” ?

- Figure 1. The difference between figures (a)-(f) is so small that I am not sure that this figure should be presented, in this form at least.

-  Figure 2. The authors write about Ag/MoO3 composite system but, as we see from figures (c) and (d), there exist separated crystalline islands of Ag and MoO3, and also separated islands of amorphous phase with oxygen vacancy defects. I think this observation requires more detailed discussion.

- The MO3 sputtering power was 50, 70 or 90 W. It is interesting whether the concentration of  MO3 is linearly proportional to the scattering power? In Figure 3, with increasing MO3 content, the Ag peak intensities become weaker. Simultaneously, most probably the MO3 peak intensities grow but how?

-  Figure 4. In the figure caption we read that UV-Vis spectra are presented but as a matter of fact these are UV-Vis-NIR spectra. The size of PS microspheres (200 nm) is comparable with the wavelength of light used in measurements of the absorption spectra. It is be expected that it could have an influence on the spectra. Have the authors considered this possibility?

-  Page 6, lines 189-190. In the Ref. [30] the Ag and Cu2S composites have been studied, i.e. not Ag/MoO3. Why this citation is given here? In Ref. [30] the LSPRs for the Ag and Cu2S composite are red-shifted nearly linearly from 580 to 686 nm. In Fig. 4 the shift seems to exist but is much smaller. In my opinion it should be discussed in more detail. What is the origin of peaks at about 350 nm and peaks marked with red stars?

-  Figure 5. Numbers on figures are too small.

In conclusion, the manuscript can be accepted for publication in Nanomaterials after a correction.

Reviewer 2 Report

The paper "Charge-transfer Induced by the Oxygen Vacancy Defects in the Ag/MoO3 Composite System" by Qi Chu et al presents the results of the systematic study of 
a Ag/MoO3 composite system sputtered on a polystyrene (PS) colloidal microsphere array.  4-Aminothiophenol (p-aminothiophenol, PATP) was used as a candidate probe molecule to evaluate the contribution of the oxygen vacancy defects in the Ag/MoO3 composite system. High-resolution TEM image analysis was used to proove the existence of oxygen vacancy defects. Surface-enhanced Raman scattering (SERS) in oxygen-containing semiconductors provides a powerful reference for the study of PATP. In particular, SERS was used to study the effects of  electron injection between the Ag/MoO3 composite and PATP. 
I think the manuscript can be published in nanomaterials after some mandatory improvements (listed below) and after the following recommendations are considered by the authors.

1) The Abstact is very labyrinthine with details. The main ideas of the manuscript  should be emphasized clearly. Authors should rewrite the Abstract to emphasize the main findings of their work. 

2) The Introduction focused mainly on the review of the composite substrates composed of, for example, noble/noble metals (Au/Ag) and semiconductors (TiO2, ZnO, Cu2O, etc.) and possible charge transfer paths in various structures. Moreover the methods of inelastic light scattering are discussed (e.g., SERS).
Since the author cite the Ref[13]-[25], it will be also relevant to touch the inelastic scattering in films on semiconductor substrates in the introduction, e.g. in [Phys. Rev. B 99, 054424. 2019], [Appl. Phys. Lett. 112, 142402 (2018)], [Phys. Rev. Applied. 9, 051002 (2018)].

3)  The authors do not explain how, how the laser power during SERS reveals the new data?  
This statement should be explained with more details since it should reflect the novelty and main findings: the overall intensity of the SERS bands decreases and the positions of the bands at 1168, 1331, and 1556cm-1 remain obvious, and their relative intensities increase, which confirms the effect of oxygen vacancy defects on the CT system

4) High-resolution TEM image analysis was used to proove the existence of oxygen vacancy defects. Authors stated that the present of an amorphous region between the two lattice planes corresponds to and indicates the existence of oxygen vacancy defects in the MoO3 structure. The citation of ref28 is presented. In ref28 we see more detailed explanation. High resolution transmission electron microscopy was used also to image an amorphous region with large size between two lattice planes. 
Then the Fourier-transform infrared reflectivity spectroscopy  was used to reveal highly accumulated oxygen vacancies in the Ar-introduced BST films. After this the conclusion about oxygen vacancies defects in the Ar-introduced BST was proposed and approved. How this can be consistent with the presented data in the current manuscript? Could one imagine that the origin of amorphous region between the two lattice planes corresponds to the existence e.g. of different phase of  MoO3. Authors should comment on this.
And ref 28 should be [Appl. Phys. Lett. 108, 033108 (2016)]

Reviewer 3 Report

The authors propose manuscript  on an interesting method/technique of Ag/MoO3 composite system on polystyrene microspheres that can be used as a SERS substrate. It has to be mentioned that the authors propose the manuscript for a publication in Nanomaterials -- the journal with 11st highest impact factor of MDPI family.  It means that authors aspire for a paper of the highest quality of MDPI publication. In my opinion, the manuscript does not achieve the quality connected to the journal. I would like to emphasize that  methods/techniques of the SERS substrate fabrication and SERS results look very promising. I see the main shortcoming of the manuscript in a complete absence of SERS experiments description. For example: one of the main parameter of SERS experiments -- excitation wavelength of Raman/SERS scattering is not clear. I have found just one small note in the caption of Fig. 5 and  still I m not sure if it is wavelength of Raman excitation (missing information about Raman spetrometer, microscope objective, power density, acquisition time, sample concentration etc). 

Also there are several minor comments: 
For example: the authors claim at the first page on lines 29-31 : 

However, noble metals  have difficulty achieving selective recognition of target molecules, are costly and have 
very low detection reproducibility; these shortcomings limit their practical applications.

Such a strong statement would certainly deserve references.